# Odor Fences Have No Effect on Wild Boar Movement and Home Range Size

**DOI:** 10.3390/ani14172556

**Published:** 2024-09-03

**Authors:** Monika Faltusová, Miloš Ježek, Richard Ševčík, Václav Silovský, Jan Cukor

**Affiliations:** 1Faculty of Forestry and Wood Sciences, Czech University of Life Sciences Prague, Kamýcká 129, 165 00 Prague, Czech Republic; jezekm@fld.czu.cz (M.J.); silovsky@lesy.czu.cz (V.S.); cukor@fld.czu.cz (J.C.); 2Forestry and Game Management Research Institute, V.V.I., Strnady 136, 252 02 Jíloviště, Czech Republic; sevcikrichard@seznam.cz

**Keywords:** GPS telemetry, crop protection, African swine fever, deterrents

## Abstract

**Simple Summary:**

The rapidly growing wild boar population has resulted in an increasing rate of human–wildlife conflicts, including economic damage to crops or spreading African swine fever (ASF), which affects the pork industry. This situation necessitates the adoption of various measures to prevent wild boar movement, mitigate spreading of diseases, and protect crops. We evaluated the impact of commonly used odor fences (Wildschwein Stopp) using GPS telemetry of tagged individuals. The telemetry of free-ranging wild boars demonstrated no effect on their crossing of the odor fence lines compared to before the installation. Moreover, no difference was found when comparing the home range size of monitored individuals during the 22 days before and after odor fence installation. Therefore, our findings do not support using odor fences to prevent wild boar movement as a mitigation measure of ASF transmission or line protection against the damage caused by wild boars to crops.

**Abstract:**

Wild boars are an opportunistic wildlife species that has successfully colonized the human-modified landscape in Europe. However, the current population boom has negative consequences, which result in a rapid increase in human–wildlife conflicts and disease transmission, including African swine fever (ASF). The increasing frequency of conflicts requires adequate solutions for these issues through various measures. Application of deterrents is a common non-lethal measure whose effects have been insufficiently verified until recently. Thus, this study aims to evaluate the effectiveness of odor fences, often applied as a barrier against wild boar movement. For this purpose, 18 wild boars were marked with GPS collars. After 22 days of initial monitoring, 12 sections of odor fences were installed on their home ranges. The monitored wild boars crossed the area 20.5 ± 9.2 times during the pre-installation period and 19.9 ± 8.4 times after the odor fence installation. Moreover, the average home range varied between 377.9 ± 185.0 ha before and 378.1 ± 142.2 ha after the odor fence installation. Based on GPS telemetry results, we do not support using odor repellent lines for crop protection or for limiting wild boar movement to lessen ASF outbreaks.

## 1. Introduction

As a species, wild boars have adapted perfectly to a human-modified landscape [1]. Changes related to the intensification of agricultural management offered wild boars an ideal environment with enough shelter and food sources for most of the year [2]. This has led to a population boom in wild boars, primarily throughout Central Europe, as the increase in their physical condition, associated with faster sexual maturation of juveniles due to available food, affects the population dynamics through earlier reproduction than previously observed. It is reflected in the rising numbers of wild boar harvested annually in countries like Spain, Poland, France, Italy, and Germany. Although growth rates varied among countries, the overall trend shows consistent population growth, with occasional stabilization periods followed by further increases [3]. However, the population increase has several explanations in Europe, including hunting regulations and hunting philosophy and the decreasing number of active hunters [3,4], very high reproduction rates [5], lack of large predators [6], reforestation, habitat alterations due to humans [7], mild winters [8], mast seeding [9], and supplementary feeding [10].

The rapid increase in the wild boar population is also associated with negative impacts on human interaction, including an increased frequency of human–wildlife conflicts [2,11]. The cost of crop damage has reached extreme amounts in certain Central European countries [12]. Still, the fundamental impacts are related to the spread of viral diseases, such as African swine fever (ASF), which is now moving from Eastern Europe through Central Europe into Western European countries [13,14]. Despite the limited host range of ASF, the socioeconomic impact of the spread of the virus is enormous [15]. Moreover, ASF outbreaks have the potential to devastate the pork industry. ASF outbreaks in China have resulted in the culling of 1.2 million pigs, and the estimated economic impact of these outbreaks is 0.78% (111.2 billion USD) of China’s gross domestic product in 2019 [16]. Similarly, ASF outbreaks in Europe caused significant declines in wild boar and domestic pig populations [17]. The assumption is that global pork prices will increase by 17–85% and the unsatisfied demand will lead to higher prices of other types of meat. For example, in 2019, beef and poultry prices rose worldwide by 1.5–6.0% and 1.6–6.7%, respectively. Of course, higher pork prices are one of the factors that reduce pork demand in all regions, with an average global per capita demand falling by 0.7–2.4 kg per year^−1^ (4–16%), with the largest consequences observed in Europe (7.9 kg per year^−1^) [18].

The increased frequency of conflicts, followed by economic impacts, has led to the use of various measures with varying effects to solve these problems. One frequently used measure is odor fences, based on a scent that simulates danger or the presence of a predator or humans [19,20,21]. Smell is a sense that serves as an extended arm of the nervous system for remote sensing of stimuli in the environment [22]. The olfactory stimulus is often one of the first impulses that alert an individual to danger. This method is used in boar population management, most often to minimize damage to agricultural and forest stands [23] or to prevent wild animal collisions on roads [24]. The odors of a natural predator [25] or an odor imitating human presence [26] are the most often used scents.

However, the effectiveness of individual measures, including odor fences, is highly debatable. In crop protection, it was evaluated as ineffective for wild boars, e.g., by Schlageter and Haag-Wackernagel, Zamojska et al. [23,27], who noted that resistance to the tested odor repellents occurred relatively quickly or was simply ineffective. Similar results were reached by Elmeros et al. [28] in cervids in forest stands. Contrarily, some studies indicate that odor repellents can be effective against browsing for up to several weeks [29,30]. The same is true in the case of reducing the number of accidents involving wild animals. Although the results show a reduction in the number of accidents after applying the so-called odor barrier by 23% to 43% [31,32], the mechanism of functionality is still not explained. The primary premise of companies that manufacture odor barriers is that installing odor fences will reduce the number of road crossings. However, [26] compared the occurrence of roe deer near roads where odor fences were installed and found no differences in the frequency of occurrence. A change in the game’s behavior rather than a reduction in the frequency of crossings is likely behind the reduced number of game collisions.

Despite the conflicting results of scientific studies, the supply of commercial preparations designed to repel wild animals from the installed fence areas is growing and is constantly finding new uses. For example, the installation of an odor barrier to prevent wild boar movement from and beyond the ASF-infected zone was utilized in the Czech Republic, Lithuania, Poland, Denmark, and India [17,33]. Because the virus did not spread from the infected area in the Czech Republic between 2017 and 2019 during the focal case [34], this method was believed to be effective. However, an exact evaluation of the reactions of the movement of wild boars to odor fences is still unexplored, and thus, it is not definitive if the odor fence was an effective measure as believed, or if the ASF virus was not spreading due to other implemented measures. As a practical matter, the only effective way to evaluate these protective measures is to use GPS-marked wild boar individuals in places with odor fences, which has not been conducted until now. 

Therefore, the aims of the presented study are (i) to evaluate the effectiveness of the application of odor fences as a barrier against wild boar movement, (ii) to assess the possible impacts of odor fences on the home ranges of GPS-marked wild boar individuals, and (iii) to evaluate any differences in behavior according to the sex of the monitored individuals.

## 2. Materials and Methods

### 2.1. Study Area

In 2019–2021, the spatial activity of wild boars was monitored at two locations. The first was the Bohumile hunting ground, Prague-East district (49.9622 N, 14.7875 E), and the second was the Hradiště hunting ground, Karlovy Vary district (50.2483 N, 13.1907 E). The Bohumile hunting ground is located in a suburban area in the wider Prague agglomeration. The area has mixed forest complexes interwoven with intensively farmed agricultural land and rural municipalities. A high level of human leisure activity is typical in the area. Contrarily, the Hradiště hunting ground is situated on the territory of a military training area, where public access is prohibited, and only activities related to forest management and army training take place here.

### 2.2. Wild Boar Telemetry

During the monitoring period (2019–2023), 62 wild boars were marked with a GPS collar (locality: 21 Hradiště, 41 Bohumile; sex: 45 females, 16 males, 1 unknown). The wild boars’ exact ages were determined by tooth eruption and then categorized into two groups: subadults (12–24 months) and adults (over 24 months). Wild boars were captured in trapping cages, immobilized, and fitted with a tracking collar [35]. The collar contained a GPS unit (Vectronic Aerospace GmBH; Berlin, Germany) and a Daily Diary biologger (Wildbyte Technologies Ltd.; Swansea, United Kingdom). We recorded data from the biologgers (3-axis accelerometer and 3-axis magnetometer with a frequency of 10 Hz). GPS positions were collected every 30 min using a GPS module and sent via SMS to an online server. We used only GPS positions with a variance of accuracy (DOP) (≥1 and ≤7) for analysis. From all the captured wild boar individuals, we included 18 wild boars in the analyses for the evaluation of the number of crossings whose movement trajectory during a 30-min interval (before the installation of the foam) crossed the planned route of the installation of the odor barrier at least five times in the control period. Other wild boars were not included in the experiment because (a) they were hunted before odor fence line application or (b) their home range and daily movement were outside the planned odor fence lines at the time of evaluation.

### 2.3. Deterrent Application

To test the effectiveness of odor fences, we used a design based on control periods [24]. The monitored period always lasted six weeks and was divided into two sections. In the control section (three weeks), no odor barrier was installed. For the experimental section, we installed a linear odor fence along the road, which we left in place for three weeks, after which the odor barrier was removed. In this study, we used the odor fence HAGOPUR—Wildschwein stop (WS-Stopp). The manufacturer (HAGOPUR AG) states that this product was developed to reduce or prevent road accidents and to prevent crop and tree browsing. Then, using an applicator, a foam was used to create the odor fence. The dispersion is a carrier material that contains a natural odor concentrate that works for one week after application. After this time, it is necessary to add concentrate for roe deer or boar (in our case, WS-Stopp). Regularly reapplying the concentrate every two to three months ensures the optimal long-term effect. According to the instructions, we applied the foam, which held to the mat, in formations roughly the size of tennis balls on the forks of branches, tree bark, tree stumps, or hammered pins. We applied the foam balls five meters from each other, as recommended by the producer, and as was evaluated before in a study by Bíl et al. [36]. According to the instructions, we added the concentrate after a week.

We installed the sections with an odor barrier based on the movement of wild boars determined by data from GPS collars (Figure 1). It means that at first, the tracked wild boar movement was evaluated for 22 days during April or October 2019–2023. Then, the line of odor fence was applied to the detected home range size, and the GPS was carried out for 22 days after application. We always placed the line of the odor fence so that it approximately intersected the area of occurrence from the last three weeks. The lines were run along public and forest roads. The length of the lines always reached a minimum of 500 m of the marked territory. Line lengths ranged from 1400 m to 3600 m. The lines were installed from April to September, i.e., when the temperature was high enough to ensure the release of the smell. We installed a total of 12 lines of odor fences and used a total of 18 wild boar individuals in the analyses that passed through the lines. The others (44) were not near the installed odor fence lines, or they missed the experiment.

### 2.4. Statistic Evaluation

We vectorized the obtained lines in QGIS 3.36 [37]. At the same time, we exported the GPS positions of the marked wild boars and selected only those that spatially and temporally corresponded to the established odor fence lines. Subsequently, we crossed the lines of the odor fences with the movement trajectories of wild boars between individual points obtained using GPS collars (interval between points, 30 min). Furthermore, we calculated the territory over which the marked individuals moved before and after the odor fence (home range) installation, using the Minimum Convex Polygon method (MCP 100%). At the same time, we exported the polygons of the home precincts and, by using the intersection of both polygons, we calculated the overlap of paired polygons of individuals before and after. The data were evaluated, visualized in R 4.2.2 software [38], tested for normality, and subsequently, evaluated for statistical differences.

We used three linear mixed-effects models (LMM) to evaluate the differences in the number of transitions, home range size, and home range overlap of wild boars (dependent variables) between the period before (A) and after (B) the installation of odor fences (independent variable) using the lmer function (lme4 and lmerTest packages) [39,40]. We included the sex (female/male) and age (adult/subadult) as covariates and locations (Bohumile/Hradiště) as random effects in all three models. The significance level was set at α = 0.05 for all statistical tests performed using the lmer function. The evaluation of differences between the periods of paired samples (identical animals monitored before and after the installation of odor fences) was also evaluated using 95% confidence intervals (CI). We checked for the normality of residuals for all models using the Shapiro–Wilk normality test and Q-Q plots.

## 3. Results

In total, we installed 12 lines of odor fences. On average, wild boars crossed the odor fence line 20.5 ± 9.2 times (mean ± SD) in the pre-installation period (A) and 19.9 ± 8.4 times when the odor fence was installed (B). The number of transitions decreased by 0.6 (95% CI, −6.0 to 4.8) after the installation of the odor barrier in the study areas. However, there is no statistical difference in the number of transitions between the two periods (Table 1, Figure 2).

The evaluated home range size before the odor fences installation (home range for 22 days) was, on average, 377.9 ± 185 ha for animals located in the areas assigned for future installation of odor fences. When the odor fences were installed, the average size of the home area was 378.1 ± 142.2 ha for the same length of the evaluated period (22 days). The average home range size decreased by 0.2 ha (95% CI, −105.6 to 106.1) after the installation of odor fences. At the same time, there was no statistical difference in the size of home ranges between periods (Table 2, Figure 3).

At the same time, there was no change in the overlap of home ranges if we compared the period before the installation and during its implementation in the study area. Before installation, the average overlap was 0.62 ± 0.17 and after, it was 0.71 ± 0.18. The average difference in overlap was 0.09 (95% CI, −0.03 to 0.20). Again, no statistical difference was noted in the overlap of home ranges between the period before and after the installation of odor fences (Table 3, Figure 4).

## 4. Discussion

Wild boar populations have been increasing worldwide in recent decades, resulting in a rapid increase in human–wildlife conflicts, such as crop damage or traffic accidents. Currently, the most negative impact is related to the spreading of ASF worldwide with negative consequences for the pork industry [41]. Therefore, various methods are being tested to channel and limit wild boar presence and movement from ASF-infected zones or valuable agricultural fields for crop protection. For these purposes, various deterrent measures are being used with limited knowledge of their effectiveness, highlighted primarily by vendors or producers claiming that the preventative substances are effective as wild boar deterrents [23].

We tested the effect of odor fences on wild boar movement in areas where the wild boar individuals were tagged with GPS telemetry transmitters, which seemed the best way to evaluate the odor fence effect, using WS-Stopp odor barriers. However, we observed no significant effect of the odor fences on wild boar movement with the installed line prepared with foam balls five meters apart. In the period before application, the wild boar crossed the line 20.5 ± 9.2 times compared to 19.8 ± 8.4 times after the fence installation. This offers new insight into evaluating the odor repellent’s effect on limiting wild boar movement in the landscape when used before to mitigate wild boar migration from ASF-affected areas or to protect attractive crops. Previously, the odor repellents were tested predominantly at baited luring sites in pairs designed where one luring site was protected and the second was the supplementary feeding site without any protection. The experiment by Schlageter and Haag-Wackernagel [23] recorded a minimal and non-significant deterrent effect of 0.4%, which means that both luring sites were visited at almost the same frequency, thus they concluded that the repellent is ineffective and not recommended for crop protection.

The monitoring of wild boar odor repellent efficiency also confirmed no effect on wild boar home range size, which was surprisingly the same for all individuals. On average, home ranges for the monitored period of 22 days were 377.9 ± 185.0 ha before and similar (378.1 ± 142.2 ha) for the period after the odor fence installation. This corresponds to monthly home range size, usually in the low hundreds of hectares [42,43,44]. For instance, in Tuscany along the Apennines, the average monthly home range size of wild boars was 187.1 ha [42], while in other Italian regions, it was 136 ha [43]. Moreover, we did not confirm significant differences in the home range sizes before and after the odor repellent installation based on the sex and age of monitored animals. This fact easily confirms that the odor fence was not respected by any age or sex group of wild boars. At the same time, no differences in the home range size of wild boar individuals according to sex classes in adult individuals are commensurate with previously published research [43,45].

There is little evidence to suggest that odor repellents effectively deter wild boar movement. Studies have shown that various types of deterrents are generally ineffective in protecting against wild boar. Benten et al. [46] revealed the ineffectiveness of wildlife warning lights in reducing wildlife–vehicle collisions on the roads. All the tested reflector models were unable to reduce the number of collisions during the experiment. Schlageter and Haag-Wackernagel [47] found LED flashers to be ineffective. In a pairing experiment, the luring sites with LED flashers were compared to those without protection. The data from 504 wild boar inspections of the luring sites indicated that solar blinkers reduced the probability of wild boar visits by 8.1% compared to the control sites. Still, the authors admit that the red light they used may have been inappropriate because wild boars seem unable to distinguish red from grey [48]. However, there are exceptions. Denzin et al. [49] reported that LED blinkers and aluminum strips performed surprisingly well in adults and juveniles, and deterrents appeared to be more effective on young wild boars. 

If the application of odor fences was to reduce the spread of diseases (including ASF), it was assumed that odor fences could be one of the solutions. As described above, our study showed that the use of odor fences is insufficient in controlling the spread of ASF. Therefore, this study does not support their use, as was previously done in the Czech Republic and Poland [50]. It appears that permanent fencing is the only effective solution to prevent the spreading of ASF, hand in hand with other measures such as reducing wild boar population, and biosecurity, which were implemented in most European countries [51]. This is true, especially for hot spot fencing used to reduce transmission of diseases once endemic, but the construction of fences requires consideration, especially in the case of wild boars [52]. The iron fence was successfully used in Belgium, eradicating a separate outbreak of African swine fever, similar to the Czech Republic. According to Mysterud and Rolandsen [52], perimeter fencing minimizes the number of animal crossings, and thus, the probability of spreading diseases. However, a problem arises with this type of fencing because of its impact on nature conservation. Furthermore, it is crucial to highlight that ASF transmission is not only due to direct contact between animals, but also through human interaction. In this case, fencing does not prevent transmission. Moreover, human disruptions can be another source that increases wild boar movement, including home range movement and long-distance dispersal [46,53], and may influence the spread of ASF. Therefore, human visitation to the affected locations, including all recreational activities, was prohibited during the ASF to prevent possible disease transmission from the outbreak in the Czech Republic. This overall restriction of access to the forest was relatively adhered to by women (who constituted only 6.7% of trespassers) compared to men (93.3%). Men and women accounted for 53.6% and 46.4% of the total visitors, respectively. Therefore, the restriction during ASF was not fully observed; consequently, the evaluation of the effectiveness of the ban on entry into the infected area is debatable [54]. The successful control measures in the Czech Republic were most likely due to muffled shootings in the high-risk ASF-outbreak areas with minimal disturbance caused by standard game management. Moreover, ostensibly, leaving attractive crops unharvested provides sufficient cover and food sources and is an effective way to mitigate wild boar movement over longer distances [34]. However, removing the carcasses and disinfecting the habitat are crucial measures in addition to those mentioned above. Altogether, the mitigation of ASF spreading is a complicated discipline made up of individual measures that can only ensure success when in sync. Based on our findings, the installation of odor fences to prevent wild boar movement between zones is not as effective as previously thought [50]. 

## 5. Conclusions

Our study investigated the efficacy of odor fences as a barrier to limit wild boar movement, which are used to control the spread of ASF and provide protection against crop damage. Despite previous theoretical support, our analysis of positioning data from GPS telemetry of free-ranging wild boars did not confirm any significant effects on the movement of tagged individuals. Wild boars crossed the odor fence lines with the same frequency after the odor fence installation as before. Moreover, we found no significant changes in their home range sizes or overlaps after the odor fence installation. This suggests that odor fences are ineffective as a short or long-term deterrent for managing wild boar populations and mitigating the spread of ASF.

Our findings concur with the preponderance of other research questioning the effectiveness of various deterrents, such as wildlife warning lights and odor repellents, in reducing wildlife-related conflicts. Although some deterrents may show short-term effects for wild boars, the evidence for these exceptions was not based on GPS telemetry. Thus, patterns describing reasons for deterrent efficiency are still unclear and can be affected by wild boar individuality or previous experience. Given the significant socioeconomic impacts of ASF and the persistent human–wildlife conflicts, exploring alternative, more reliable methods is crucial. Permanent perimeter fencing, intensive surveillance, and strategic hunting practices appear more effective. However, these solutions also negatively affect wildlife, including non-targeted species, which needs to be considered before installation. Therefore, future efforts should focus on developing and testing new deterrent systems that consider the wild boar’s natural behavior and movement patterns and effectively address this ongoing issue.

## Figures and Tables

**Figure 1 animals-14-02556-f001:**
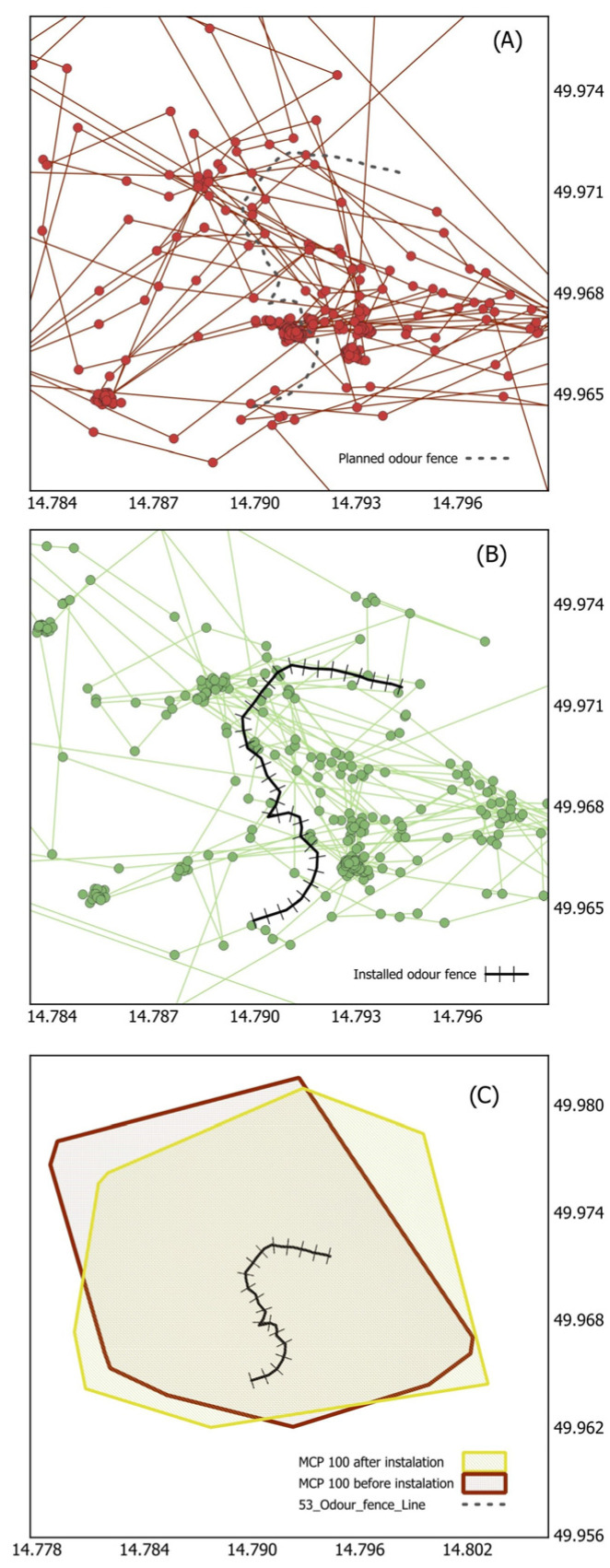
The movement of a wild boar during the tested period depicted as connections between individual GPS positions (30-min intervals) before the installation of the odor fence (**A**) and after its installation (**B**); overlap of the utilized area (MCP 100%) before and after the installation of the odor fence (**C**).

**Figure 2 animals-14-02556-f002:**
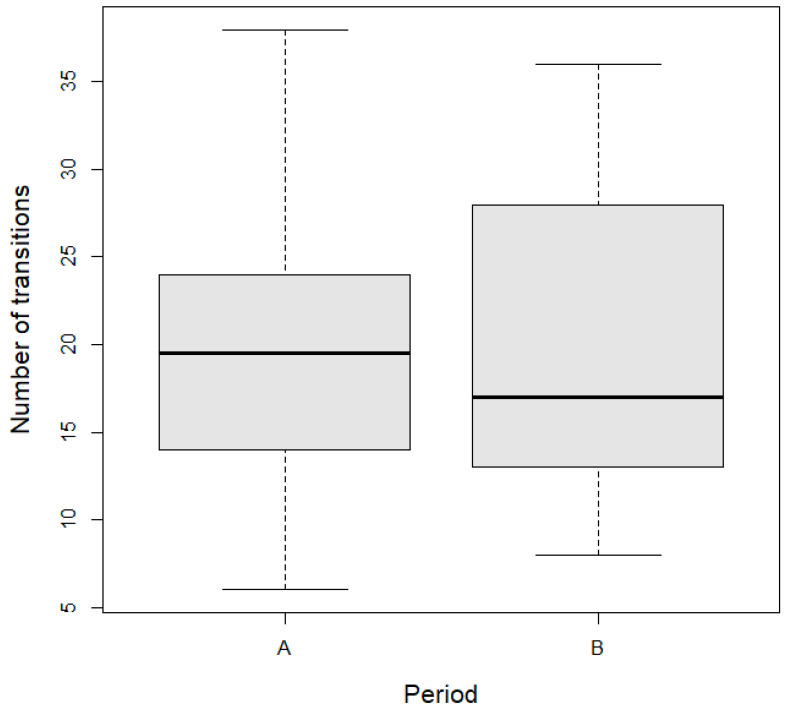
Number of wild boar transitions before (A) and after (B) installation of odor fences in the study area. Boxplots show the median values (middle bar in rectangles), upper and lower quartiles (length of rectangles), and maximum and minimum values (whiskers).

**Figure 3 animals-14-02556-f003:**
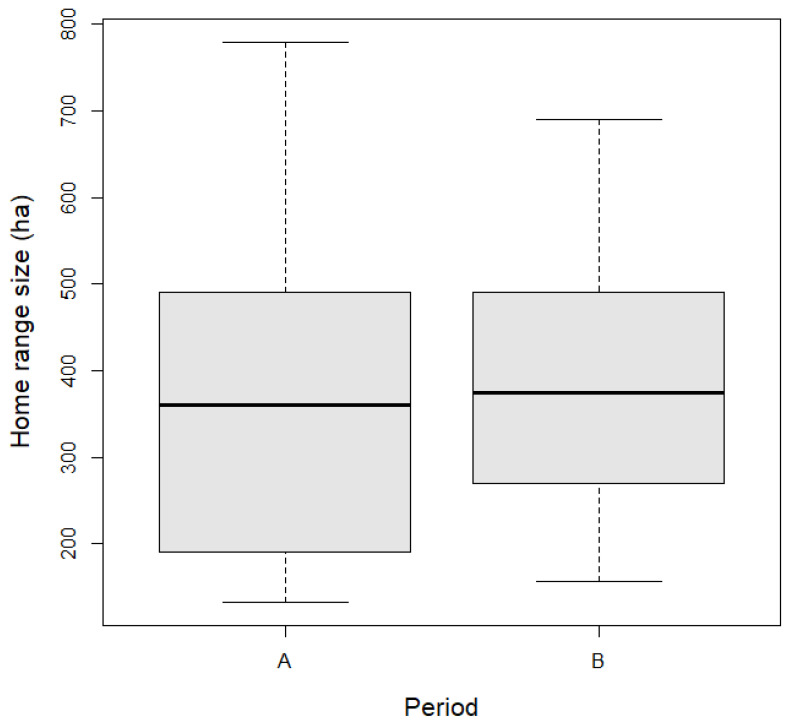
Average size of home range during two periods, before (A) and after (B) installation of odor fences in the study area. Boxplots show the median values (middle bar in rectangles), upper and lower quartiles (length of rectangles), and maximum and minimum values (whiskers).

**Figure 4 animals-14-02556-f004:**
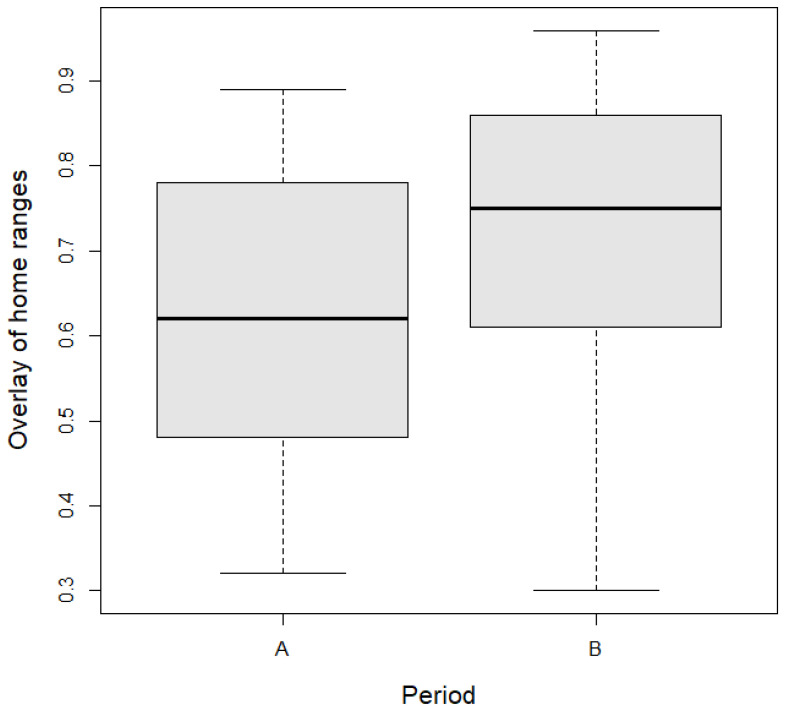
Overlay of home ranges before (A) and after installation (B) of odor fences in the study area. Boxplots show the median values (middle bar in rectangles), upper and lower quartiles (length of rectangles), and maximum and minimum values (whiskers).

**Table 1 animals-14-02556-t001:** Results of the linear mixed-effects model testing differences between the number of transitions (dependent variable) and periods (independent variable) in the study areas. Sex and age were included as covariates and locations as random effects. For each variable, we report the slope (estimate) and its standard error (SE), t-values, and *p*-values.

Variable	Estimate	SE	t-Value	*p*-Value
Intercept	23.6748	4.4175	5.359	0.0526
Period (B)	−0.6111	2.8010	−0.218	0.8287
Sex (M)	−1.0040	3.3941	−0.296	0.7693
Age (S)	−2.8906	3.0233	−0.956	0.3464

**Table 2 animals-14-02556-t002:** Results of the linear mixed-effects model testing differences between the home range size (dependent variable) and periods (independent variable) in the study areas. Sex and age were included as covariates and locations as random effects. For each variable, we report the slope (estimate) and its standard error (SE), t-values, and *p*-values.

Variable	Estimate	SE	t-Value	*p*-Value
Intercept	388.5824	57.0172	6.815	0.0083
Period (B)	0.2222	55.0948	0.004	0.9968
Sex (M)	−72.0198	66.5263	−1.083	0.2872
Age (S)	35.0190	59.4157	0.589	0.5599

**Table 3 animals-14-02556-t003:** Results of the linear mixed-effects model testing the differences between home range overlap (dependent variable) and periods (independent variable) in the study areas. Sex and age were included as covariates and locations as random effects. For each variable, we report the slope (estimate) and its standard error (SE), t-values, and *p*-values.

Variable	Estimate	SE	t-Value	*p*-Value
Intercept	0.6528	0.0641	10.191	0.0033
Period (B)	0.0861	0.0601	1.433	0.1619
Sex (M)	−0.0048	0.0726	0.066	0.9478
Age (S)	−0.0442	0.0648	−0.682	0.5002

## Data Availability

The original data used for the study are included in the Appendix A; further inquiries can be directed to the corresponding author.

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
