# Peer review of "Odor Fences Have No Effect on Wild Boar Movement and Home Range Size"

_animals, 2024, doi:10.3390/ani14172556_

Round 1

Reviewer 1 Report

Comments and Suggestions for Authors

       Line 43: However, the population increase has several explanations in Europe, including hunting regulations and ethics and the decreasing number of active  hunters [3,4]. It is unclear how ethics affects boar population – this could use additional context.

       Line 56: Similarly, wild boar outbreaks in Europe caused significant population declines. Outbreaks of ASF in wild boars? Declines in the population of what (domestic pigs, wild boars)?

       Line 58: For example, in 2019, beef and poultry 58 prices rose by 1.5–6.0% and 1.6–6.7%, respectively. Where did these price increases occur? Some likely are in the range for overall inflation.

       Line 59: Meat consumption has dropped for other reasons beyond the price of pork.

       Methods: How many boars were tracked at any given time? Were animals used unique? How were animals trapped? How long was telemetry collected? Was sex undetermined in one animal? It’s unclear how the deterrent was used. Other investigators (Schlageter and Haag-Wackernagel, 2012) describe aluminum strips with felt pads – is this similar to the current product? How application standardized beyond stating @ the size of a tennis ball. What type of mat? Spacing of the mat? Was prevailing wind direction considered? Any effort to evaluate whether the odor deterrent works under controlled conditions? Indicate that the material was applied per manufacturer’s recommendations. Number of relevant subjects = 18 – elsewhere n = 62 is mentioned. When were GPS collars removed? Was the commercially available product used within any product expiration date? How did you determine that the 5 m spacing could work? Demographics of the 18 subjects was not provided. Not clear when the fence was created relative to when GPS monitoring occurred (e.g., the fence may have been initially created in April and testing occurred months later so animal might become acclimated to the deterrent).

       Line 155: Statistical methods. Clearly state which cohort of animals was evaluated (I assume n = 18 rather than 62). Clearly indicate the inclusion criteria used to identify relevant subjects (it’s unclear why 48 animals were not used). How were animals uniquely identified (e.g., GPS collar). How was age determined (e.g., dentition).

       Figure 3 averages appear different. The Figures are duplicating information provided in the text and are not needed.

       A major concern is a lack of statistical power – how much of a change would be considered important (e.g., 20% decrease in range? 20% decrease in odor fence crosses). No attempt to define a useful outcome has been made nor has there been any discussion of statistical powered considerations

Comments on the Quality of English Language

Well written manuscript.

Author Response

Dear reviewer,

thank you for your valuable feedback on our research. We have revised the statistical section of our work, which we believe has greatly enhanced the manuscript. The results are now presented more clearly for readers. We have addressed all your suggestions in detail below.

Comments: Line 43: However, the population increase has several explanations in Europe, including hunting regulations and ethics and the decreasing number of active hunters [3,4]. It is unclear how ethics affects boar population – this could use additional context.

Response: Thank you very much for your comment. We agree with you. We changed text to „hunting philosophy“ which means that for example, hunters in central Europe do not hunt adult wild boar females at all…

Comments: Line 56: Similarly, wild boar outbreaks in Europe caused significant population declines. Outbreaks of ASF in wild boars? Declines in the population of what (domestic pigs, wild boars)?

Response: I rewrote it more clearly, thanks for the warning.

Comments: Line 58: For example, in 2019, beef and poultry 58 prices rose by 1.5–6.0% and 1.6–6.7%, respectively. Where did these price increases occur? Some likely are in the range for overall inflation.

Response: Thank you for this comment, the prices increased worldwide which we have clearly mentioned in the text now.

Comments: Line 59: Meat consumption has dropped for other reasons beyond the price of pork.

Response: I added that it is one of the factors.

Comments: Methods: How many boars were tracked at any given time? Were animals used unique? How were animals trapped? How long was telemetry collected? Was sex undetermined in one animal? It’s unclear how the deterrent was used. Other investigators (Schlageter and Haag-Wackernagel, 2012) describe aluminum strips with felt pads – is this similar to the current product? How application standardized beyond stating @ the size of a tennis ball. What type of mat? Spacing of the mat? Was prevailing wind direction considered? Any effort to evaluate whether the odor deterrent works under controlled conditions? Indicate that the material was applied per manufacturer’s recommendations. Number of relevant subjects = 18 – elsewhere n = 62 is mentioned. When were GPS collars removed? Was the commercially available product used within any product expiration date? How did you determine that the 5 m spacing could work? Demographics of the 18 subjects was not provided. Not clear when the fence was created relative to when GPS monitoring occurred (e.g., the fence may have been initially created in April and testing occurred months later so animal might become acclimated to the deterrent).

Response: In total, we tracked 62 wild boar, but only 18 of them tested the odour fence; other animals were not included in the experiment because of time of tracking (winter time – when the odour fence is not working, and when the smell does not spread because of low temperature) or the animals lived in the areas where was not able to install the odour fence (because of terrain), or the animals were shot/collar stopped worked during the time of the experiment or the animals were simply not in the area with odour fence (during the experiment).

Yes, each animal was part of the test only once – the animals in the results are unique.

Animals were captured in trapping cages installed in the hunting grounds and immobilized.

The monitored period always lasted six weeks and was divided into two sections. In the control section (three weeks), no odor barrier was installed. For the experimental section, we installed a linear odor fence along the road, which we left in place for three weeks, after which the odor barrier was removed.

Yes, in one animal, we could not detect the sex because of the impaired function of anesthesia during collaring, and the animal escaped during the handling.

Thank you for this comment. We used the methodology which is described by the producers (Hagopur), and which is described by other publication Bíl et al. 2024 "Olfactory repellents decrease the number of ungulate-vehicle collisions on roads: Results of a two-year carcass study." Journal of Environmental Management 365 (2024): 121561.

We did not used aluminum strips because the aim of the study was to evaluate only the effect of odour repellents not the aggregate effect of detergent combination. Therefore, we used the methodology according to product producer as we clearly mentioned now in the methodology.

Most of the animals were shot during the experiment, which was designed in standard hunting grounds and, therefore, in standard conditions of game management. The average tracking period for a total of 62 animals was 121 days. Then, the collar was removed by drop-of function, or, in most cases, the individual was hunted, as we now mentioned in the manuscript. 

The 5 meters is recommended by the producers (see https://www.hagopur.de/produktwelt/premium-schutzmittel/wildunfalle/). Therefore, we tried to monitor the possible effect of odour fence in real situation as it is applied in praxis. All odour tubes were used before the expiration date. 

The boar were monitored at two locations, the location was not statistically significant and therefore not included in the analyses.

The fences were installed between April and October. In total, we installed 12 different line transects. There was no possibility that the animal might become acclimated to the deterrent because monitoring the odour fence's effect was begun immediately after application. We have now clearly described this process in the manuscript; thank you very much for this comment.

Comments: Line 155: Statistical methods. Clearly state which cohort of animals was evaluated (I assume n = 18 rather than 62). Clearly indicate the inclusion criteria used to identify relevant subjects (it’s unclear why 48 animals were not used). How were animals uniquely identified (e.g., GPS collar). How was age determined (e.g., dentition).

Response: We used only 18 individuals because Line 148: We installed a total of 12 lines of odor fences and used a total of 18 wild boar individuals in the analyzes that passed through the lines. The others (44) were not near the installed odor fence lines, or they missed the experiment.

Line 115 answers part of your note. A total of 62 individuals were tagged using GPS collars. The collar contained a GPS unit (Vectronic Aerospace GmBH) and a Daily Diary biologger (Wildbyte Technologies Ltd). The exact age was determined by the tooth eruption and the animals were divided to the subadults (12-24 months) and adults (24 months+).

Comments: Figure 3 averages appear different. The Figures are duplicating information provided in the text and are not needed.

Response: Figure 3 does not show average values but median values, so figures do not duplicate information provided in the text. More values, such as median, max, min or upper and lower quartiles, are not mentioned in the text. In these types of studies, boxplots are typically used to visualize data distributions, where the median is shown along with the interquartile range. The boxplot effectively highlights the central tendency and variability of the data, using the median as a key measure of central tendency. In addition to the boxplot representation, we have also reported the mean and standard deviation in the text (line 210-216). While the boxplot emphasizes the median, the inclusion of the mean and standard deviation provides additional insight into the data’s overall distribution, allowing for a more comprehensive understanding of the central tendencies and variability.

Comments: A major concern is a lack of statistical power – how much of a change would be considered important (e.g., 20% decrease in range? 20% decrease in odor fence crosses). No attempt to define a useful outcome has been made nor has there been any discussion of statistical powered considerations.

Response: Thank you very much for the note, which we considered and therefore decided to rerun the analysis with more straightforward methods. We used lmer models with covariates (sex and age) and random effects (locations) to account for both fixed and random sources of variability in our data (line 179). Including covariates in the model allows us to control for potential confounding factors that might influence the relationship between main predictor (period) and the outcome. This ensures that the effects we observe are more accurately attributed to the variables of interest, rather than to other unrelated factors. Subsequently, the residuals of the models were tested for normality using the Shapiro-Wilk test and qqplot, which confirmed that the residuals were normally distributed. This validation allows us to use this model to describe the average differences in home range size, number of transitions, and home range overlap between periods (before and after the installation of the odor fences). However, none of the observed differences between the two periods were statistically significant, which was our primary goal of the study, to determine and describe from our data whether odor fences affect movement of wild boars. To provide an overview and description of the data we obtained, we included the differences between periods as determined by the models and the 95% confidence intervals in the text.

Best regards,

Monika Faltusová

Reviewer 2 Report

Comments and Suggestions for Authors

The ms is worth publishing after minor changes and implementations. You should better explain the main causes of the dramatic increase of wild boar in Central Europe, quoting the specific literature. It would be also useful to add some examples of the demographic increment in the last decade for a few countries (estimates of population size or harvest statistics before ASP; see Apollonio et al. 2010 “European Ungulates and their Management…”; Massei et al. 2015). For the mean home range size, it would be better to quote other examples from Central Europe. The final paragraph on literature should be corrected and implemented.

I add some notes on specific points:

Lines 43-45 : Please be more precise on the several causes of the wild boar demographic increase of the last five decades. You should include at least the climate warming, with warmer winters and consequently greater juvenile survival (cf Vetter et al. 2015 “What is a mild winter…”) and with enhanced seeding rate of oaks and beeches in autumn (cf Keuling et al. 2018 “Eurasian wild boar Sus scrofa” in Melletti & Meijaard eds. “Ecology, conservation and management of wild pigs and peccaries”; Touzot et al. 2020 “How does increasing mast seeding…”). You should also quote supplementary feeding and reforestation.

Line 255: in another area of Tuscany

Please, add other examples of monthly home ranges for wild boar.

Line 310 : Please cancel from the text “In conclusion” at the beginning of a paragraph entitled Conclusion.

Line 364: 2019 in bold fonts

Line 391 (but also lines 397, 402, 421): Name of the genus with a capital letter, name of the species with a lowercase letter.

393: Please cancel the second 2020

408: 2018 in bold fonts

423: Cavazza, S.; Brogi, R.; Apollonio, M.

443: names of the authors of the quoted paper in lowercase letters.

Comments on the Quality of English Language

The English seems sufficiently correct, but a critical reading by a native speaker would be useful.

Author Response

Dear reviewer,

Thank you for your insightful feedback on our research. Following your recommendation, we have added additional information to strengthen our introduction. We have addressed all your suggestions in detail below.

Comments: The ms is worth publishing after minor changes and implementations. You should better explain the main causes of the dramatic increase of wild boar in Central Europe, quoting the specific literature. It would be also useful to add some examples of the demographic increment in the last decade for a few countries (estimates of population size or harvest statistics before ASP; see Apollonio et al. 2010 “European Ungulates and their Management…”; Massei et al. 2015). For the mean home range size, it would be better to quote other examples from Central Europe. The final paragraph on literature should be corrected and implemented.

Response: We added a more general sentence. We did not give specific cases for randomly selected countries.

Comments: I add some notes on specific points:

Lines 43-45: Please be more precise on the several causes of the wild boar demographic increase of the last five decades. You should include at least the climate warming, with warmer winters and consequently greater juvenile survival (cf Vetter et al. 2015 “What is a mild winter…”) and with enhanced seeding rate of oaks and beeches in autumn (cf Keuling et al. 2018 “Eurasian wild boar Sus scrofa” in Melletti & Meijaard eds. “Ecology, conservation and management of wild pigs and peccaries”; Touzot et al. 2020 “How does increasing mast seeding…”). You should also quote supplementary feeding and reforestation.

Response: Thanks for the note, we have edited it in the manuscript.

Comments: Line 255: in another area of Tuscany

Please, add other examples of monthly home ranges for wild boar.

Response: Unfortunately, we were unsuccessful with it during writing, and this literature is unavailable. The known literature regarding home ranges was either daily, seasonal, or annual. Due to the nature of the project, we decided on a monthly one.

Comments: Line 310: Please cancel from the text “In conclusion” at the beginning of a paragraph entitled Conclusion.

Response: Thank you, we deleted it.

Comments: Line 364: 2019 in bold fonts

Response: Thank you, changed.

Comments: Line 391 (but also lines 397, 402, 421): Name of the genus with a capital letter, name of the species with a lowercase letter.

Response: Thank you, we did it.

Comments: 393: Please cancel the second 2020

Response: Thank you, we deleted it.

Comments: 408: 2018 in bold fonts

Response: Thank you, changed.

Comments: 423: Cavazza, S.; Brogi, R.; Apollonio, M.

Response: Thank you, changed.

Comments: 443: names of the authors of the quoted paper in lowercase letters.

Response: Thank you, changed.

Best regards,

Monika Faltusová

Round 2

Reviewer 1 Report

Comments and Suggestions for Authors

Update number of subjects in the abstract - currently suggests n = 62

Author Response

Dear reviewer,

we appreciate your valuable feedback on our research. Based on your recommendation, we have update abstract.

 Comments: Update number of subjects in the abstract - currently suggests n = 62

Response: Thank you, we updated the number n = 18.